# Galangal–Cinnamon Spice Mixture Blocks the Coronavirus Infection Pathway through Inhibition of SARS-CoV-2 M^Pro^, Three HCoV-229E Targets; Quantum-Chemical Calculations Support In Vitro Evaluation

**DOI:** 10.3390/ph16101378

**Published:** 2023-09-28

**Authors:** Doaa G. El-Hosari, Wesam M. Hussein, Marwa O. Elgendy, Sara O. Elgendy, Ahmed R. N. Ibrahim, Alzhraa M. Fahmy, Afnan Hassan, Fatma Alzahraa Mokhtar, Modather F. Hussein, Mohamed E. A. Abdelrahim, Eman G. Haggag

**Affiliations:** 1Pharmacognosy Department, Faculty of Pharmacy, Helwan University, Cairo 11795, Egypt; ph.wesam95@gmail.com (W.M.H.); eman.g.haggag@pharm.helwan.edu.eg (E.G.H.); 2Department of Clinical Pharmacy, Beni-Suef University Hospitals, Faculty of Medicine, Beni-Suef University, Beni-Suef 62521, Egypt; 3Department of Clinical Pharmacy, Faculty of Pharmacy, Nahda University (NUB), Beni-Suef 62513, Egypt; 4Clinical and Chemical Pathology Department, Faculty of Medicine, Beni-Suef University, Beni-Suef 62521, Egypt; drsaraosama88@med.bsu.edu.eg; 5Department of Clinical Pharmacy, College of Pharmacy, King Khalid University, Abha 61421, Saudi Arabia; aribrahim@kku.edu.sa; 6Tropical Medicine and Infectious Diseases Department, Faculty of Medicine, Beni-Suef University, Beni-Suef 62521, Egypt; alzhraamohamed@med.bsu.edu.eg; 7Biomedical Sciences Program, University of Science and Technology, Zewail City of Science and Technology, Cairo 12578, Egypt; t-afhassan@zewailcity.edu.eg; 8Department of Pharmacognosy, Faculty of Pharmacy, El Saleheya El Gadida University, El Saleheya El Gadida 44813, Egypt; fatma.mokhtar@sgu.edu.eg; 9Chemistry Department, Collage of Science, Jouf University, P.O. Box 2014, Sakaka 72388, Saudi Arabia; mfhussin@ju.edu.sa; 10Chemistry Department, Faculty of Science, Al-Azhar University, Asyut Branch, Assiut 71524, Egypt; 11Clinical Pharmacy Department, Faculty of Pharmacy, Beni-Suef University, Beni-Suef 62521, Egypt; mohamed.abdelrahim@pharm.bsu.edu.eg

**Keywords:** galangal–cinnamon aqueous extract, SARS-CoV-2 M^Pro^, antiviral assay, phenolic compounds, quantum-chemical methods

## Abstract

Natural products such as domestic herbal drugs which are easily accessible and cost-effective can be used as a complementary treatment in mild and moderate COVID-19 cases. This study aimed to detect and describe the efficiency of phenolics detected in the galangal–cinnamon mixture in the inhibition of SARS-CoV-2’s different protein targets. The potential antiviral effect of galangal–cinnamon aqueous extract (GCAE) against Low Pathogenic HCoV-229E was assessed using cytopathic effect inhibition assay and the crystal violet method. Low Pathogenic HCoV-229E was used as it is safer for in vitro laboratory experimentation and due to the conformation and the binding pockets similarity between HCoV-229E and SARS-CoV-2 M^Pro^. The GCAE showed a significant antiviral effect against HCoV-229E (IC_50_ 15.083 µg/mL). Twelve phenolic compounds were detected in the extract with ellagic, cinnamic, and gallic acids being the major identified phenolic acids, while rutin was the major identified flavonoid glycoside. Quantum-chemical calculations were made to find molecular properties using the DFT/B3LYP method with 6-311++G(2d,2p) basis set. Quantum-chemical values such as EHOMO, ELUMO, energy gap, ionization potential, chemical hardness, softness, and electronegativity values were calculated and discussed. Phenolic compounds detected by HPLC-DAD-UV in the GCAE were docked into the active site of 3 HCoV-229E targets (PDB IDs. 2ZU2, 6U7G, 7VN9, and 6WTT) to find the potential inhibitors that block the Coronavirus infection pathways from quantum and docking data for these compounds. There are good adaptations between the theoretical and experimental results showing that rutin has the highest activity against Low Pathogenic HCoV-229E in the GCAE extract.

## 1. Introduction

The whole world has faced the coronavirus pandemic. This virus has infected millions of people and killed thousands [1]. It causes severe respiratory issues and multi-organ failure, where the mortality rate increases if the patient has a comorbid illness or is elderly [2,3]. At present, we do not have any available and affordable specific treatment for COVID-19. Consequently, the easiest way to avoid coronavirus is to prevent the possible causes and boost the body’s immune response by eating nutritious meals with proper dietary consumption, such as fruits, vegetables, proteins, spices, and nuts [4,5].

Spices can be used to make herbal drinks that also serve as medicinal drinks, as they contain phytoconstituents that can minimize cell damage and reduce inflammation [6]. Because of the simplicity and ease of serving instant drinks, such as domestic herbal tea bags, they are currently in high demand among the general public. Recently, using herbal plants is favorable due to their fewer side effects and natural origin compared to drugs of chemical origin; they have anti-inflammatory, antipyretic, and antimicrobial properties, and they have secondary metabolites such as polyphenols, flavonoids, tannins, alkaloids, lignins, coumarins, stilbenes, and terpenoids [7,8]. Flavors and sweeteners could be added to herbal tea drinks to improve their taste and make them more appealing. As a result, we have decided to assess the consequences of using herbal tea in coronavirus patients with moderate illness and explore its effectiveness.

The application of virtual computational studies is considered the first obligatory step in the drug discovery field. The computational tools are time-saving, economically valuable, and are of high need in highly infectious diseases. The computational studies include molecular docking [9], molecular dynamics [10], quantum studies [11], and pharmacology networking [12]. These tools are able to discover new medical applications of existing drugs or natural products. The determination of possible pathways or involved genes can be performed through these virtual computational tools [13].

In this study, we selected galangal and cinnamon to be used as a mixture. According to recent studies, these two herbs are among some herbal plants that exert an antiviral effect on the Coronavirus family members [14]. The galangal (*Alpinia officinarum* Hance) (lesser galangal) rhizome belongs to the family Zingiberaceae, and cinnamon (*Cinnamomum verum* J.Presl) (Ceylon cinnamon) bark belongs to the family Lauraceae. In silico investigations have stated that these two herbs can inhibit the activity of Coronaviruses as they have various effective constituents such as flavonoids, essential oils, and terpenoids [15,16]. This research aims to evaluate the inhibitory and antiviral efficacy of a galangal–cinnamon mixture on the Coronavirus family and discover which of its major phenolic constituents contribute to these effects.

## 2. Results

### 2.1. Antiviral Effect of the Galangal–Cinnamon Aqueous Extract (GCAE) on Low Pathogenic Coronavirus (229E)

The galangal–cinnamon aqueous extract (GCAE) was tested to examine its inhibitory effect on Low Pathogenic Coronavirus HCoV-229E compared to remdesivir as a standard antiviral drug, using the cytopathic effect (CPE) inhibition assay and the crystal violet method [17,18]. The antiviral activities of the extract and the remdesivir standard were determined using a concentration range from 0.1 to 1000 µg/mL (10-fold dilutions). The development of the cytopathic effect was monitored by light microscopy and the 50% inhibitory concentration (IC_50_) results were determined for both (see Table 1 and Figure 1).

### 2.2. HPLC-UV Analysis of Phenolic Compounds in GCAE, Galangal Aqueous Extract (GAE), and Cinnamon Aqueous Extract (CAE)

HPLC-DAD-UV analysis was performed for the galangal–cinnamon extract (GCAE) and for cinnamon (CAE) and galangal (GAE) extracts individually at the same conditions against 19 phenolic standards to evaluate the effect of admixing and the major flavonoids and phenolic acids in the galangal–cinnamon mixture (Table 2, Figure 2). HPLC-DAD chromatograms of cinnamon and galangal extracts individually are illustrated in Figure 3

### 2.3. Quantum-Chemical Calculations

Quantum-chemical calculations of the major detected phenolics in GCAE were made to find molecular properties. Quantum-chemical values such as EHOMO, ELUMO, energy gap (∆E), ionization potential (I), chemical hardness (η), softness (σ), electronegativity (X), and hydrophobicity were calculated and listed in Table 3 using density functional theory (DFT), which predicts the molecular reactivity descriptors by Equations (1)–(8). Frontier molecular orbitals (HOMOs, LUMOs) of the GCAE detected phenolic compounds and the molecular electrostatic potential (MEP) surface are represented in Figure 4.

The frontier molecular orbitals (FMO) of the detected phenolic compounds show that ELUMO and EHOMO represent the capacity of a compound to receive or donate electrons, respectively. DFT-based parameters have been calculated to estimate the reactivity of the detected compounds, where the reactivities of these compounds are arranged according to their activity. The highest active molecule was rutin with the lowest ∆E value and ionization potential. It is noted in this table that the energy gap between EHOMO and ELUMO of most bioactive molecules varies between 0.1 to 0.17, except for rutin and chlorogenic acid, which have higher activity of 0.00965 and 0.02206, respectively, and this is caused by the presence of a large number of (OH) phenolic groups in these two molecules. The hardness values of the phenolic compounds were calculated and are presented in Table 3; these range from 0.01 to 0.07 eV, except rutin, which has a value of 0.0048 eV, and this means that rutin is the softest and most active compound. The softness values of most phenolic compounds vary between 13 and 18 eV, except rutin and chlorogenic acid, which have the highest values. Overall, the activity of the galangal–cinnamon spice mixture can be predicted by its softness value, which is a whole number. Therefore, the activity of the mixture can be predicted by the following: Firstly, by collecting the experimental concentrations of each and dividing each one of them by the total, then multiplying by resulting the percentage of each component in relation to the total of 12 detected phenolic compounds. The next step involves multiplying each of these values by its softness value to produce a valuable effectiveness, then the effectiveness of each component is divided by the total to produce the quantum relative concentration. According to the hypothesis, softness values, and experimental concentrations, rutin shows the highest quantum relative concentration (61.95%), compared to the other phenolic compounds detected in the galangal–cinnamon aqueous extract. The data presented in Table 3 show a high reactivity characteristic of bioactive compounds that resulted from the presence of (OH) phenolic groups, which have a lone pair of electrons that act as a nucleophile and can interact with soft acceptor molecules (electrophile) as the coronavirus protein.

The EHOMO and ELUMO maps of the major molecules are shown in Figure 4 and the rest of the molecules are presented in Appendix A. The ELUMO and EHOMO map shows anti-bonding and bonding characteristics, from which it can be seen that the electron densities of the FMOs are more concentrated on the six-membered ring and attached by OH groups. These regions are assumed to be chemically active, in accordance with the frontier molecular orbital theory.

Electrostatic potential (ESP) suggests the molecules’ electrophilic and nucleophilic nature, and it is an important tool to study the compounds reactivity nature. The maps of ESP at the surface are expressed by 17 distinct colors. The blue color stands for the highest amount of the positive region where the nucleophilic reaction occurs, and the reddish region indicates the negative region where the electrophilic reaction takes place, while zero potential is represented by the green color. The molecular electrostatic potential (ESP) of the bioactive molecules is shown in Figure 4 and Appendix A. Most of the electron density is localized on the OH phenolic groups (active site), which act as nucleophiles. The ESPs show that the oxygen atoms of the hydroxyl group provide favorable sites for hard–hard interactions (Figure 4 and Appendix A) between the nucleophile and the electrophile, which are active sites for electrophilic attacks. Accordingly, rutin has negative electrostatic potential as the highest active molecule compared to the others.

Quantum-chemical methods play an essential role in the molecular reactivity or stability determination. Reported researches revealed that FMO theory is effective in predicting interactive centers of molecules [19]. The methods used depend on measuring HOMO–LUMO energy gap (∆E) and MEP and using DFT calculations [20]. MEP is utilized to outline the electrostatic interaction between a molecule and an atom.

### 2.4. Molecular Docking Studies

To find the potential target for GCAE phenolic extracts, a molecular docking study was carried out on three HCoV-229E targets: M^pro^, RBD and spike glycoprotein, in addition to SARS-CoV-2 M^pro^. The predicted binding energies of the 12 phenolic compounds in addition to remdesivir (positive control) on 2ZU2, 6U7G, 7VN9, and 6WTT active pockets are listed in Table 4. Rutin possesses the highest binding affinity among all GCAE extracts against the three targets, confirming our findings in quantum-chemical calculations that rutin is the most reactive compound.

Rutin was able to bind with HCoV-229E M^pro^ with a ΔG of −7.6667 Kcal/mol, which is approaching remdesivir’s binding affinity (−8.7640 Kcal/mol). Moreover, rutin was able to form a large number of interactions with the M^pro^ pocket including nine H-bonds, a pi–pi, pi–anion, and pi–alkyl interactions with 10 different residues, as shown in Figure 5. Some of these residues are the same ones which interact with remdesivir, like Gly 142, ILE 164, and GLU 165. In addition to these residues, rutin was able to interact with M^pro^ catalytic dyad residues (HIS41/CYS144), two main reported residues for M^pro^ inhibition [21]. The remaining 11 phenolic extracts also reported good binding affinities but less than rutin towards inhibiting M^pro^ (−4.2583 < ΔG < −5.9897) (Table 4 and Appendix A).

For HCoV-229E RBD Class V, rutin reported a better binding score towards its active pocket than remdesivir’s results (ΔG = −7.0607 and −6.7442, respectively) (Table 4). It also binds to the RBD pocket with four H-bonds and two pi–pi stacking interactions, which is better than remdesivir, which forms only two H-bonds with RBD active residues (Appendix A). Likewise is the case with spike glycoprotein: Rutin was able to bind to spike glycoprotein with a very stable binding energy (−6.7486 Kcal/mol). It interacts with the same residues as remdesivir, like TYR 354, TYR 406, and ARG 350 (Appendix A). Like M^pro^, all other GCAE phenolic extracts were able to block RBD and spike glycoprotein but with lower affinities (Appendix A). The second-best phenolic compound that was able to bind to the three targets is chlorogenic acid. This also confirms QC calculations where chlorogenic acid is reported to be the second most soft phenolic derivative due to the presence of a large number of (OH) groups.

The galangal–cinnamon spice mixture was also tested for its capability to block the Coronavirus infection pathway through the inhibition of SARS-CoV-2 M^pro^, where they were docked against PDB ID 6WTT. The binding energies shown in Table 4 present similar energies for all compounds to those of HCoV-229E M^pro^. This brings us to the 2D interactions of the best hit (rutin) and positive control (remdesivir) on the SARS-CoV-2 M^pro^ pocket and compare them to those of HCoV-229E M^pro^ (Figure 5). As shown in Figure 5, the key interactions are reserved in the four complexes, especially the catalytic dyad HIS 41/CYS 145, where rutin was also able to form five H-bonds with SARS-CoV-2 M^pro^ in addition to pi–pi stacking interaction. These very similar binding affinities and interactions prove our postulate that, despite the low (48%) homologs identity between SARS-CoV-2 and HCoV-229E (M^pro^), the active pockets of both are of identical sequence, and hence, any detected hit on the less virulent one can also inhibit the SARS-CoV-2 protease.

## 3. Discussion

The COVID-19 pandemic is a serious global health concern, and finding new and effective ways to deal with its risks is critical. Natural products have become potential therapeutic alternatives for treating several diseases, such as viral infections, due to their innately high tolerance in the human body [22]. Galangal is very useful for its uses in food and medicine. It can be used for treating hemorrhoids, abdominal discomfort, abnormal menstruation, inflammation, metastatic breast cancer, and as an anti-aging agent [23]. Cinnamon has been used in cooking since centuries past and has a wide variety of culinary applications. It is a widely used spice that has been proven to be helpful in promoting health, owing to its specific properties. These characteristics could be beneficial in the treatment of a vast range of diseases, including cardiovascular disease, cancer, diabetes, and neurological issues. Cinnamon’s properties and antioxidant activities are responsible for its medicinal benefits [24].

The galangal–cinnamon aqueous extract (GCAE) was tested to examine its inhibitory effect on Low Pathogenic Coronavirus HCoV-229E compared to remdesivir as a standard antiviral drug, using the cytopathic effect (CPE) inhibition assay and the crystal violet method. Given that the SARS-CoV-2 virus is fatal and cannot be assessed safely and easily in the laboratories, we chose the less pathogenic HCoV-229E to assess the extract’s inhibitory activity. The identity of SARS-CoV-2’s and HCoV-229E’s main protease (M^pro^) homologs is just about 48% according to the amino acid sequence alignment, with the binding sites being almost identical. This similarity remains true with respect to the conformation and binding pocket of the M^pro^ enzyme in both viruses [25]. The GCAE showed significant antiviral activity with IC_50_ equal to 15.083 µg/mL, almost half the antiviral potency of remdesivir (IC_50_, 8.76 µg/mL). Remdesivir is an antiviral drug with potent broad-spectrum in vitro and in vivo antiviral activity against the COVID family [26,27].

HPLC-DAD-UV analysis of (GCAE), (CAE), and (GAE) revealed that ellagic, cinnamic, and gallic acids are the major identified phenolic acids in GCAE in concentrations of 4057.01, 1789.61, 871.34 µg/g, respectively, while rutin is the major flavonoid glycoside identified in a concentration of 1293.35 µg/g. Other phenolic compounds were not detected in GCAE, although they were detected in GAE or CAE individually. This may refer to the mixing effect, which could oxidize or reduce or form other chemical modifications such as chelation or precipitation due to compound–compound interactions and the mixture preparation technique.

We build the hypothesis at the expense of calculating the quantum relative concentration for each detected phenolic compound of the GCAE. Collecting the experimental concentrations of each compound and dividing each one of them by the total, then multiplying by 100, results in the percentage of each component in relation to the total of 12 detected phenolics. Multiplying each of these values by its softness value produces a valuable effectiveness, then the effectiveness of each component is divided by the total to produce the quantum relative concentration. According to the hypothesis, softness values, and experimental concentrations, rutin shows the highest quantum relative concentration (61.95%) compared to the other phenolic compounds detected in the galangal–cinnamon aqueous extract (Table 5).

Molecular docking was applied to predict the specific target of these detected phenolic compounds in the GCAE. Virtual screening on three potential targets HCoV-229E revealed that rutin is the main component responsible for GCAE inhibitory activity. Rutin possesses a very high affinity towards inhibiting M^pro^, spike glycoprotein, and RBD [28,29]. These three main targets are essential for the entry and replications of SARS-CoV-2 via RBD blockage inhibiting RBD-hACE2 interaction and also inhibiting the viral main protease, which is involved in the viral replication. Therefore, identifying a natural product that can block these three targets at the same time is a win–win situation.

Phenolic compounds in natural medicinal plants offer various beneficial health effects such as antioxidant [30], anti-microbial [31], anti-inflammatory [32], as well as anti-carcinogenic effects. In addition to their proven role here as significant antivirals by the in silico studies carried out, phenolic compounds that are prominent in GCAE have been proved in the literature to exert other activities useful to assess COVID patients in recovery. Ellagic acid has antioxidant and anti-inflammatory effects that are considered as cofactors in the speed recovery of COVID patients [33]. Moreover, cinnamic acid modulates inflammatory cytokines in adipose tissue, the liver, the hypothalamus, and decreases TNF-α in serum [34]. Gallic acid is known for its strong free radical scavenging activities and anti-inflammatory properties; it has also shown immunomodulatory effects, as supported by a decrease in the inflammation cytokines and an increase of anti-inflammatory cytokines [35]. Additionally reported data stated that rutin has an immune regulatory effect, which can regulate cytokines and immune-related signaling pathways and cause a strong enhancement of antibody levels, increase immunoglobulin levels, and restore the functioning of leucocytes [36].

## 4. Materials and Methods

### 4.1. Collection of Medicinal Plants

Dried rhizomes of galangal and barks of cinnamon were purchased from a local spice shop in Cairo, Egypt, and identified by Prof. Loutfy M. Hassan, professor of Flora and Plant Ecology, Botany Department, Faculty of Science, Helwan University, Cairo, Egypt. A sample of each herb was assigned a voucher number, 28-Aof-3/2021 for galangal and 36-Cve-1/2021 for cinnamon, and kept in the herbarium of the Faculty of Pharmacy, Pharmacognosy Department, Helwan University, Cairo, Egypt. Powders of each sample were prepared by grinding to a suitable particle size and were stored in glass jars at room temperature in a dark, dry place until investigation. Pharmacognostic and microbiological quality control tests were carried out for the powders for safety and quality assurance.

### 4.2. Preparation of the Herbal Extracts

For the preparation of the galangal–cinnamon mixture aqueous extract, we followed the optimized easy-to-use at-home extraction protocol [37] to imitate the effect of domestic herbal mixture infusion. The herbal mixture (2.5 g galangal + 2.5 g cinnamon powders) was emptied into a 500 mL flask and extracted with de-ionized boiling water (250 mL), mixed well by manual rotation for 5 s, covered, and allowed to brew for 6–10 min. It was then filtered with Whatman No.1 filter paper, and the filtrate was concentrated and lyophilized to obtain the dry crude extract (325 mg). The galangal–cinnamon aqueous extract (GCAE) was stored in the dark at 4 °C for further investigation. Using the same procedure, galangal and cinnamon extracts were prepared separately starting with 2.5 g powder for each, yielding 117 mg galangal (GAE) and 93 mg cinnamon (CAE) aqueous extracts.

### 4.3. Antiviral Assay Using Low Pathogenic Coronavirus (229E)

The cytopathic effect (CPE) inhibition assay and the crystal violet method were applied to pinpoint the potential antiviral effect of GCAE against Low Pathogenic Coronavirus (229E) compared to remdesivir as a standard antiviral drug [38]. Coronavirus (229E) and Vero E6 cells were offered by Nawah-Scientific, Egypt. Vero E6 cells were grown in DMEM medium supplemented with 10% fetal bovine serum (Grand Island, NY, USA) and 0.1% antibiotic/antimycotic solution (Gibco BRL Thermo Fisher Scientific Inc., Waltham, MA, USA). The method, in brief, is as follows: Vero E6 cells at a density of 2 × 10^4^ cells/well were bedded into a 96-well culture plate one day before infection. The next day, the culture medium was eliminated, and the cells were rinsed with phosphate-buffered saline (PBS). The infectivity of the Coronavirus was determined by employing the crystal violet method, which screened CPE and helped in the calculation of the percentage of cell viability. Virus suspension 229E (0.1 mL) containing cell culture infectious dose 50% CCID 50 (1.0 × 10^4^) of virus stock was put on Vero E6 cells, as this selected dose gave the desired CPEs after two days of infection. Mediums (0.01 mL) each containing the GCAE and remdesivir concentrations were added to the cells. The antiviral activity of each sample was determined using a concentration range of 0.1–1000 µg/mL (10-fold dilutions). Control cells were used in the experiment (virus-infected, non-drug-treated cells) for virus controls and non-infected, non-drug-treated cells for cell controls. For 72 h, the culture plates were incubated at 37 °C in 5% carbon dioxide. The development of the CPE was tracked by light microscopy. After washing with PBS, the cell monolayers were fixed and stained with a 0.03% crystal violet solution in 2% ethanol and 10% formalin. The optical density of each well, after washing and drying, was quantified spectrophotometrically at 570/630 nm. The samples’ antiviral activities percentage was calculated according to Pauwels et al., 1988 [18] using the following equation:
Antiviral activity = [MOD of cell controls − MOD of virus controls(MOD sample − MOD of virus controls)] × 100%.
MOD = Mean Optical Density.

The 50% CPE inhibitory dose (IC_50_) was calculated based on these results. The 50% inhibitory concentration (IC_50_) results were determined using Graph pad prism software version 8 (Graph-Pad Software, San Diego, CA, USA).

### 4.4. High-Performance Liquid Chromatography–Ultraviolet Analysis (HPLC-UV)

Phenolic compounds determination of the GCAE and of GAE and CAE separately were analyzed by the HPLC-DAD-UV technique, and separation was carried out on Eclipse C18 column on Agilent 1260 instrument. Phenolic compounds standards were purchased from Sigma Aldrich Merck, Darmstadt, Germany, Merck. Water (A) and 0.05% trifluoroacetic acid in acetonitrile (B) constituted the mobile phase at a flow rate of 0.9 mL/min. The elution was performed as follows: 0 min (82% A); 0–5 min. (80% A); 5–8 min. (60% A); 8–12 min. (60% A); 12–15 min. (82% A); 15–16 min. (82% A); and 16–20 (82%A) in a linear gradient. The multi-wavelength detector was monitored at 280 nm using a Diod array detector. The volume of injection for each of the sample solutions was 5 μL, and the column temperature was maintained at 40 °C.

### 4.5. Quantum-Chemical Calculations

Quantum-chemical calculations of the detected compounds were made to find molecular properties using the Gauss View 06 and Gaussian 09 program package [39,40]. EHOMO and ELUMO analysis, molecular structure, and MEP (Molecular Electrostatic Potential) of detected compounds were determined by using Becke–3–Lee Yang Parr (B3LYP) [41] using the DFT/B3LYP method with 6-311++G(d,p) basis set, and their 3D plots were verified with density functional theory (DFT) methods with 6-311++G(d,p) in the ground state. From the data obtained from the gauss view, the theoretical values for EHOMO, ELUMO, ∆E, electron affinity, ionization potential, softness, hardness, electronegativity, and hydrophobicity were calculated according to Oyewole et al., 2020 [42]. The quantum-chemical descriptors were calculated using Equations (1)–(8). Both physical and chemical properties of the molecules are intimately interrelated to the energy gap (∆E, Equation (1)) [43,44]. One of the essential parameters of chemical reactivity is the ionization potential (Equation (2)). A high value of ionization potential demonstrates strong stability and low chemical activity, while a low value predicts the high activity of compounds [43]. In chemistry, the terms “hardness (η)” and “softness (σ)” are frequently employed to describe the stability of molecules. The concept of chemical hardness (Equation (3)), first used by Pearson in the 1960s, is the resistance of chemical species to electron cloud deformation or polarization [45]. Therefore, chemical hardness could be used to predict the stability of the molecules. A compound is referred to as hard if it has a significant energy gap, and as soft otherwise [46]. Accordingly, active molecules possess high softness and low hardness values (Equation (4)). Further, the chemical potential (µ) was calculated according to Equation (5). The electronegativity (χ) is considered as the negative value of µ (Equation (6)). In general, a compound with decreased electronegativity possesses a robust ability to donate electrons, resulting in more activity than a compound with a high electronegativity value. The susceptibility of the compound to accept electrons or electron density is indicated by the electronegativity as well [47] (Equation (7)). The electron affinity (A) can be considered as the negative value of E_LUMO_ [48]. Another important reactivity indicator for comparing compounds’ potential to donate electrons is the global electrophilicity indicator (ω, Equation (8)) [49]. A strong electrophile has a significant electrophilicity value, whereas a strong nucleophile has a low electrophilicity value [50].

Equations:∆E = ELUMO − EHOMO(1)
I = −EHOMO(2)
η = (1 − A)/2(3)
σ = 1/2η(4)
µ = −χ ≅ (EHOMO + ELUMO)/2(5)
χ = (I + A)/2(6)
A = −ELUMO(7)
ω = μ2/2η(8)

All methods were carried out in accordance with relevant guidelines.

### 4.6. Molecular Docking Studies

Twelve phenolic compounds detected by HPLC-DAD-UV in the GCAE were tested in silico against 3 different targets of HCoV-229E in addition to the SARS-CoV-2 main protease to find out a potent target for them. Four crystal structures were chosen and procured from the PDB (Protein Data Bank), https://www.rcsb.org/, namely (accessed on 10 March 2023), PDB ID. 2ZU2 for HCoV-229E M^pro^, PDB ID. 6U7G for HCoV-229E RBD Class V, PDB ID. 7VN9 for HCoV-229E spike protein receptor-binding domain, and PDB ID. 6WTT for SARS-CoV-2 M^pro^. Only the needed chains were kept and the receptor structures were prepared using Auto Dock Vina, where polar hydrogens were added and energy was minimized utilizing the prepare_receptor4.py command of the ADT. The receptors pockets were chosen for docking according to those reported in literature [51,52].

For the phenolic compound’s chemical structures, a test set of the previously minimized structured obtained from QM step using the DFT was used and converted to PDBQT format via AutoDockTools (ADT, v1.5.6). Moreover, remdesivir was docked on the same target pockets as a positive control. The search engine used was the Lamarckian genetic algorithm with local search, with 100 runs and a population size of 150. To evaluate the docking results, ten conformers of the ligands were considered. Eventually, the conformer with the minimal binding free energy was evaluated and 2D interaction figures were generated via BIOVA Discovery Studio visualizer 2021.

## 5. Conclusions

The whole world has faced the Coronavirus pandemic, where hospitals were filled with critical cases and mild cases were sent home with the government routine protocol therapy and were remotely monitored. This study was designed to evaluate the effectiveness of some domestic herbs on COVID-19 and shed the light on the myth of whether they actually help or not. The galangal–cinnamon aqueous extract exerts a significant antiviral effect against HCoV-229E with IC_50_ equal to 15.083 µg/mL, almost half the antiviral potency of remdesivir. Among the twelve phenolic compounds detected in the mixture’s aqueous extract, it was found that rutin showed the highest activity against HCoV-229E M^pro^, RBD, spike glycoproteins, and SARS-CoV-2 M^pro^ when tested in silico.

From these findings, we conclude that this mixture with the same ratio can offer a domestic herbal tea to help mild COVID-19 patients to safely manage their manifestations.

## Figures and Tables

**Figure 1 pharmaceuticals-16-01378-f001:**
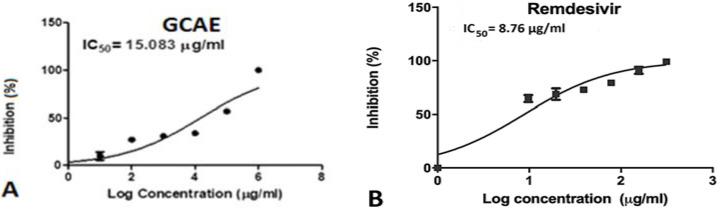
Inhibitory effects of GCAE (**A**) and remdesivir (**B**) on HCoV-229E. The IC_50_ value for each is inserted in each plot.

**Figure 2 pharmaceuticals-16-01378-f002:**
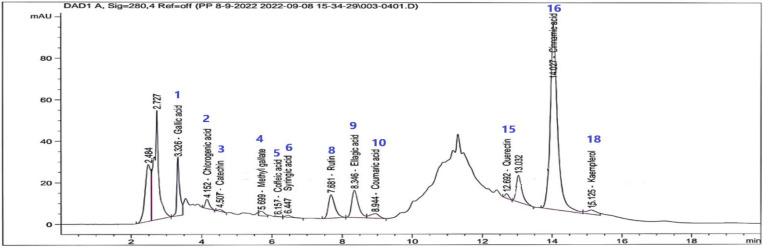
HPLC-DAD chromatogram of GCAE.

**Figure 3 pharmaceuticals-16-01378-f003:**
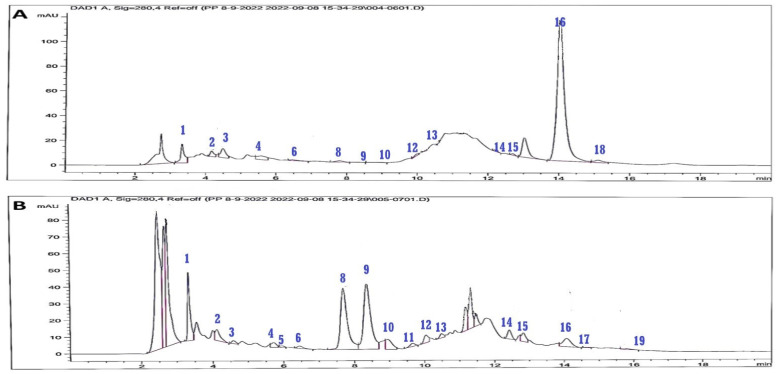
HPLC-DAD chromatograms of (**A**) CAE, (**B**) GAE.

**Figure 4 pharmaceuticals-16-01378-f004:**
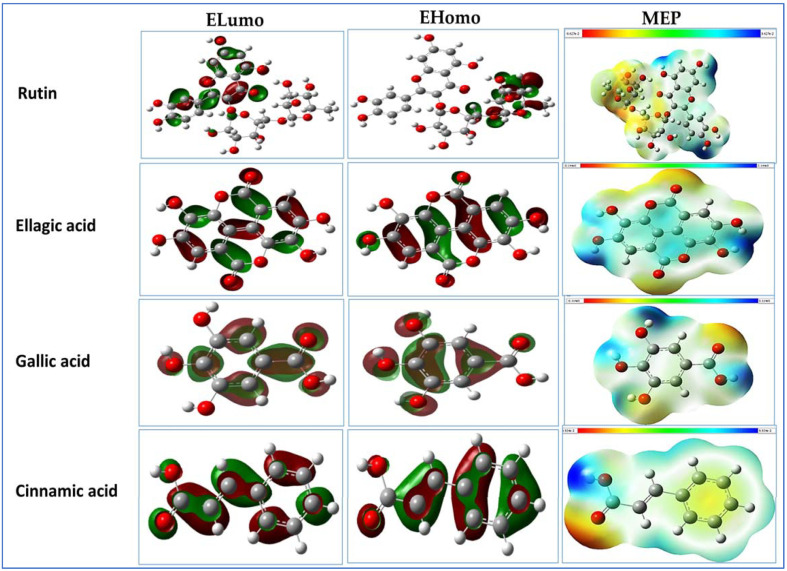
Frontier molecular orbitals (HOMOs, LUMOs) and molecular electrostatic potential (MEP) surface of the rutin, ellagic, gallic, and cinnamic acids by using the DFT/B3LYP/6-311++G(2d,2p) method.

**Figure 5 pharmaceuticals-16-01378-f005:**
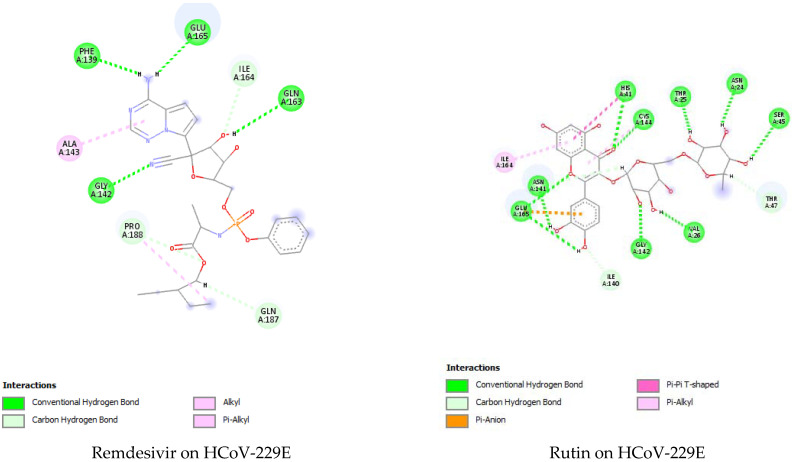
Two-dimensional interactions rutin (best hit) and remdesivir (positive control) docked on HCoV-229E and SARS-CoV-2 main protease.

**Table 1 pharmaceuticals-16-01378-t001:** The antiviral activity of GCAE compared to remdesivir against HcoV-229E.

	The 50% Inhibitory Concentration (IC_50_) µg/mL
GCAE	15.083
Remdesivir	8.76

**Table 2 pharmaceuticals-16-01378-t002:** HPLC-DAD analysis results of GCAE compared to GAE and CAE.

Cpd No.	RT (min)	Concentration (µg/g)			
		Phenolic Compounds	GCAE	CAE	GAE
1	3.32	Gallic acid	871.34	570.91	607.15
2	4.131	Chlorogenic acid	386.20	338.86	366.10
3	4.51	Catechin	140.10	1145.88	133.45
4	5.98	Methyl gallate	94.78	202.50	61.67
5	5.92	Coffeic acid	6.26	0.00	23.61
6	6.46	Syringic acid	68.35	51.77	48.43
7	6.67	Pyro catechol	0.00	0.00	0.00
8	7.76	Rutin	1293.35	123.13	2442.36
9	8.62	Ellagic acid	4057.01	64.08	7244.25
10	8.95	Coumaric acid	79.59	7.86	80.72
11	9.63	Vanillin	0.00	0.00	24.00
12	10.14	Ferulic acid	0.00	45.14	82.58
13	10.43	Naringenin	0.00	164.60	66.57
14	12.23	Daidzein	0.00	29.76	107.54
15	12.72	Quercetin	153.88	72.55	215.67
16	14.01	Cinnamic acid	1789.61	2153.55	63.39
17	14.49	Apigenin	0.00	0.00	7.77
18	15.00	Kaempferol	454.72	352.05	0.00
19	15.58	Hesperetin	0.00	0.00	20.81

**Table 3 pharmaceuticals-16-01378-t003:** Calculated quantum-chemical parameters of the detected phenolic compounds using the B3LYP/6-311++G(2d,2p) method.

Phenolic Cpds	ELUMO	EHOMO	ΔE	A	I	X	η	σ	ω
Rutin	−0.18509	−0.19474	0.00965	0.18509	0.19474	0.189915	0.004825	207.2539	7.475172
Chlorogenic acid	−0.17956	−0.20162	0.02206	0.17956	0.20162	0.19059	0.01103	90.66183	3.29325
Quercetin	−0.16462	−0.27037	0.10575	0.16462	0.27037	0.217495	0.052875	18.91253	0.89464
Kaempferol	−0.16468	−0.27157	0.10689	0.16468	0.27157	0.218125	0.053445	18.71082	0.890233
Coffeic acid	−0.17679	−0.28741	0.11062	0.17679	0.28741	0.2321	0.05531	18.07991	0.973972
Ellagic acid	−0.16152	−0.27310	0.11158	0.16152	0.2731	0.21731	0.05579	17.92436	0.846453
Coumaric acid	−0.17712	−0.29902	0.1219	0.17712	0.29902	0.23807	0.06095	16.40689	0.929899
Syringic acid	−0.15418	−0.28435	0.13017	0.15418	0.28435	0.219265	0.065085	15.36452	0.738682
Cinnamic acid	−0.18398	−0.31745	0.13347	0.18398	0.31745	0.250715	0.066735	14.98464	0.941905
Gallic acid	−0.15421	−0.28885	0.13464	0.15421	0.28885	0.22153	0.06732	14.85443	0.728989
Methyl gallate	−0.15393	−0.28880	0.13487	0.15393	0.2888	0.221365	0.067435	14.82909	0.726662
Catechin	−0.13643	−0.28986	0.15343	0.13643	0.28986	0.213145	0.076715	13.03526	0.592202

ELUMO energy of the lowest unoccupied molecular orbital, EHOMO energy of the highest occupied molecular orbital, ∆E HOMO–LUMO energy gap, A electron affinity, I ionization potential, X electronegativity, η chemical hardness, σ chemical softness, ω global electrophilicity.

**Table 4 pharmaceuticals-16-01378-t004:** Docking ΔG scores of 12 phenolic compounds and remdesivir on 2ZU2, 6U7G, 7VN9, and 6WTT active pockets.

Cpd	ΔG (Kcal/mol)			
	2ZU2	6U7G	7VN9	6WTT
Remdesivir	−8.7640	−6.7442	−6.8270	−8.5391
Rutin	−7.6667	−7.0607	−6.7486	−8.5519
Chlorogenic acid	−5.9897	−5.3078	−5.0056	−6.8119
Quercetin	−5.7422	−5.1478	−4.9061	−6.1569
Kaempferol	−5.5863	−4.9378	−4.9439	−6.1614
Caffeic acid	−4.4261	−4.2180	−4.1875	−4.8618
Ellagic acid	−5.1831	−4.7615	−4.7619	−5.9172
Coumaric acid	−4.4789	−4.1064	−4.2298	−4.7411
Syringic acid	−4.7306	−4.4892	−4.6226	−5.1851
Cinnamic acid	−4.2583	−4.1707	−4.1572	−4.6575
Gallic acid	−4.3674	−3.9848	−4.0322	−4.5764
Methyl gallate	−4.4899	−4.2326	−4.3924	−4.8474
Catechin	−5.6074	−5.1583	−5.1000	−6.1628

**Table 5 pharmaceuticals-16-01378-t005:** The calculated activity of each detected phenolic compound in the GCAE using the softness value.

	Chemical Softness (σ)	Experimental Concentration µg/g	Experimental Concentration %	Activity	Activity %
Rutin	207.2539	1293.35	13.76609	2853.07515	61.95381
Chlorogenic acid	90.66183	386.2	4.110614	372.675792	8.092561
Quercetin	18.91253	153.88	1.6378 59	30.9760645	0.672637
Kaempferol	18.71082	454.72	4.839923	90.5589357	1.966464
Coffeic acid	18.07991	6.26	0.06663	1.2046615	0.026159
Ellagic acid	17.92436	4057.01	43.18178	774.005717	16.80734
Coumaric acid	16.40689	79.59	0.847136	13.8988607	0.30181
Syringic acid	15.36452	68.35	0.7275	11.1776871	0.242721
Cinnamic acid	14.98464	1789.61	19.04815	285.429689	6.198034
Gallic acid	14.85443	871.34	9.27432	137.76474	2.991527
Methyl gallate	14.82909	94.78	1.008814	14.9597949	0.324848
Catechin	13.03526	140.1	1.491189	19.4380308	0.422092

## Data Availability

Data are contained within the article and Appendix A.

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
