# Peer review of "Galangal–Cinnamon Spice Mixture Blocks the Coronavirus Infection Pathway through Inhibition of SARS-CoV-2 M^Pro^, Three HCoV-229E Targets; Quantum-Chemical Calculations Support In Vitro Evaluation"

_pharmaceuticals, 2023, doi:10.3390/ph16101378_

Round 1
Reviewer 1 Report
It is very difficult to conclude that GCAE is a potent compound based on several experiments shown in this study. However, it is worth reporting the weak bioactivity of the GCAE compound. Additionally, it is challenging to claim that all compounds found in GCAE directly bind to the main protease. The inhibitory effects comparison in Table 1 is not clear enough to suggest that the target of GCAE is similar to Remdesivir. Therefore, this work should perform pan-docking for all HCoV-229E non-structural proteins, not just the main protease, and compare them to the reported inhibitors for each protein target.
Moreover, using SARS-CoV-2 main protease (6WTT) as the protein receptor for molecular docking in this study is not appropriate due to the low sequence similarity and difficulty in accepting it. The authors should consider performing homology modeling or other techniques to construct the HCoV-229E main protease and use that for molecular docking. This would help highlight the potent target for GCAE.
The ligand used in the molecular docking was constructed and minimized from ChemDraw using MM forcefield. Using the optimized ligand structure from QM would be better using the DFT/B3LYP method with 6-311++G(d,p) instead of Chemdraw for molecular docking. The manuscript lacks a discussion on molecular docking and the QM parts.
Figures 2-3: Please remake them to improve resolution and quality.
Figure 4: This figure should only show key compounds (Ellagic acid, Rutin, Cinnamic acid, Gallic acid) based on the HPLC-DAD analysis. The others should be moved to the Supporting Information (SI). Labels in 2D and 3D should be enlarged. The current version's surface binding is less informative and should also be moved to the SI.
Figure 5: This figure should only show key compounds, and the others should be moved to the SI. "Mol." should be changed to "cpd" for consistency. Labels in 2D and 3D should be enlarged and made clearer. "basic set" appears to be a typo. All figures here have a disproportionate appearance, with atoms appearing oval-shaped. Please remake them. The ESP surface lacks a spectrum bar.
Line 109: "phenolic compounds' molecules" - change "'s" to a general style. (Please edit "'s" throughout the manuscript)
Line 110 and 120: "6Wtt protein" is a typo. "COVID-19 Mpro" should be changed to "SARS-CoV-2 Mpro" (Please edit throughout the manuscript)
Line 111: "energy scores (S)" should be changed to "binding energy scores (S)"
Lines 181-191: This explanation should be moved to the Results section.
Line 296: "Gauss View 06" - remove the space. (Please edit throughout the manuscript)
Line 299: Please add a citation for B3LYP.
Line 302: Please add the definition of ∆E here.
Lines 301-302: Recapitalize the font.
Lines 325-333: Please use the MDPI format for equations.
Line 339: "herps" appears to be a typo.
Line 341: "IC50" is not consistent in format.
Author Response
It is very difficult to conclude that GCAE is a potent compound based on several experiments shown in this study.
However, it is worth reporting the weak bioactivity of the GCAE compound.
Additionally, it is challenging to claim that all compounds found in GCAE directly bind to the main protease. The inhibitory effects comparison in Table 1 is not clear enough to suggest that the target of GCAE is similar to Remdesivir.
Docking study was redone to identify the remdesivir targets and the GCAE phenolic targets to determine the similarity
Therefore, this work should perform pan-docking for all HCoV-229E non-structural proteins, not just the main protease, and compare them to the reported inhibitors for each protein target. Moreover, using SARS-CoV-2 main protease (6WTT) as the protein receptor for molecular docking in this study is not appropriate due to the low sequence similarity and difficulty in accepting it. The authors should consider performing homology modeling or other techniques to construct the HCoV-229E main protease and use that for molecular docking. This would help highlight the potent target for GCAE.
The whole docking study was done again, where all the compounds were tested in silico against 3 different targets of HCoV-229E in addition to SARS-CoV-2 main protease to find out a potent target for them. Four crystal structures were chosen and procured from The PDB (Protein Data Bank), https://www.rcsb.org/.; PDB ID. 2ZU2 for HCoV-229E Mpro, PDB ID. 6U7G for HCoV-229E RBD Class V, PDB ID. 7VN9 for HCoV-229E spike protein receptor-binding domain and PDB ID. 6WTT for SARS-CoV-2 Mpro.
The ligand used in the molecular docking was constructed and minimized from ChemDraw using MM forcefield. Using the optimized ligand structure from QM would be better using the DFT/B3LYP method with 6-311++G(d,p) instead of Chemdraw for molecular docking.
We have repeated the docking procedures after minimizing the test set using DFT as suggested
The manuscript lacks a discussion on molecular docking and the QM parts.
A new better discussion was written
Comments on the Quality of English Language
Figures 2-3: Please remake them to improve resolution and quality.
Figures resolution were improved
Figure 4: This figure should only show key compounds (Ellagic acid, Rutin, Cinnamic acid, Gallic acid) based on the HPLC-DAD analysis. The others should be moved to the Supporting Information (SI). Labels in 2D and 3D should be enlarged. The current version's surface binding is less informative and should also be moved to the SI.
Docking study was repeated and all the related figures were moved to supplementary file except only a figure describing the highest binding affinity compounds to different proteins
Figure 5: This figure should only show key compounds, and the others should be moved to the SI. "Mol." should be changed to "cpd" for consistency. Labels in 2D and 3D should be enlarged and made clearer. "basic set" appears to be a typo. All figures here have a disproportionate appearance, with atoms appearing oval-shaped. Please remake them. The ESP surface lacks a spectrum bar.
Docking study was repeated and all the related figures were moved to supplementary file except only a figure describing the highest binding affinity compounds to different proteins
. "Mol." Is changed to "cpd"
All figures appearance are adjusted
Line 109: "phenolic compounds' molecules" - change "'s" to a general style. (Please edit "'s" throughout the manuscript)
All the s mistypes are corrected
Line 110 and 120: "6Wtt protein" is a typo. "COVID-19 Mpro" should be changed to "SARS-CoV-2 Mpro" (Please edit throughout the manuscript)
CORRECTED
Line 111: "energy scores (S)" should be changed to "binding energy scores (S)"
Removed at all
Lines 181-191: This explanation should be moved to the Results section.
The explanation is removed to the results
Line 296: "Gauss View 06" - remove the space. (Please edit throughout the manuscript)
Corrected to GaussView 6.0
Line 299: Please add a citation for B3LYP.
Citation is added
Line 302: Please add the definition of ∆E here.
∆E+ energy gap ( it is described where its first mention located)
Lines 301-302: Recapitalize the font.
The font is capitalized
Lines 325-333: Please use the MDPI format for equations.
MDPI equations format is applied
Line 339: "herps" appears to be a typo.
corrected
Line 341: "IC50" is not consistent in format.
corrected
Reviewer 2 Report
Overall Comments
The authors presented a manuscript entitled “Galangal-cinnamon spice mixture block the Coronavirus infection pathway through inhibition of SARS-CoV-2 MPro, Quantum-chemical calculations supports in vitro evaluation” in which antiviral activities, chemistry computing and analytical chemistry methods were employed in the evaluation of the study object components’ antiviral activity against SARS-CoV-2. The English is good and well written. The methodology employed is scientifically sound. The text can improve a bit in the discussion, I felt that all the employed methods could improve in the discussion and be better tied together. Also, an increased discussion of the probable action mechanisms of this antiviral activity would enrich the article. Nevertheless, the findings are good and the article is an interesting read.
Specific Comments
Abstract: “Phenolic compounds detected by HPLC-DAD-UV in the GCAE were docked into the active site of 6wtt to find the potential inhibitors that block the Coronavirus infection pathway through inhibition of SARS-CoV-2 MPro using Quantum-chemical calculations which were made to find molecular properties using DFT/B3LYP method with 6-311++G(2d,2p) basis set” In this text the authors meant that the docking was performed and latter some molecular properties were calculated with DFT, is that correct? The way this section is phrased make it seem like the docking employed the DFT calculation
Quantum data is too vague, please specify
In figure 1, what is the scale of the point in the horizontal axis? It doesn’t seem to be logarithmic scale; Moreover, 1-B has a (0,0) point?
The choice to compare chemical softness calculated by DFT with the experimental activity was interesting and does make sense. However, is not clearly discussed how the chemical softness of this phenolic compounds were related to the proposed antiviral activity.
The docking calculations seemed not to play a relevant role in the article’s findings. The docking can corroborate the findings of the manuscript, but it is not clearly written.
Author Response
Comments and Suggestions for Authors
Overall Comments
The authors presented a manuscript entitled “Galangal-cinnamon spice mixture block the Coronavirus infection pathway through inhibition of SARS-CoV-2 MPro, Quantum-chemical calculations supports in vitro evaluation” in which antiviral activities, chemistry computing and analytical chemistry methods were employed in the evaluation of the study object components’ antiviral activity against SARS-CoV-2. The English is good and well written. The methodology employed is scientifically sound. The text can improve a bit in the discussion, (the whole section was updated)
I felt that all the employed methods could improve in the discussion and be better tied together. Also, an increased discussion of the probable action mechanisms of this antiviral activity would enrich the article. Nevertheless, the findings are good and the article is an interesting read.
Specific Comments
Abstract: “Phenolic compounds detected by HPLC-DAD-UV in the GCAE were docked into the active site of 6wtt to find the potential inhibitors that block the Coronavirus infection pathway through inhibition of SARS-CoV-2 MPro using Quantum-chemical calculations which were made to find molecular properties using DFT/B3LYP method with 6-311++G(2d,2p) basis set” In this text the authors meant that the docking was performed and latter some molecular properties were calculated with DFT, is that correct? The way this section is phrased make it seem like the docking employed the DFT calculation
We rephrased this section
Quantum data is too vague, please specify
Quantum chemical calculations can be used to predict the biological activity of compounds by calculating their molecular properties, such as the energy gap between the HOMO and LUMO orbitals, electron affinity, ionization potential, hardness, and softness. These properties can be used to identify compounds that are likely to interact with biological targets. The extracted active molecules of the Galangal-cinnamon spice mixture were evaluated in table 3 using DFT that predicts the molecular reactivity descriptors by equation (1-8). Generally, a compound with a small energy gap between the HOMO and LUMO orbitals is likely to be a good electron donor, while a compound with a large energy gap is likely to be a good electron acceptor. It is noted in this table that the energy gap between E Homo and Lumo of most bioactive molecules varies between 0.1 to 0.17, except for Rutin and Chlorogenic acid, which have higher activity of 0.00965 and 0.02206, and this is caused by the presence of a large number of (OH) phenolic groups in these two molecules. On the other hand, compound with a high electron affinity is likely to be a good nucleophile, while a compound with a high ionization potential is likely to be a good electrophile. A soft compound is one that has a large electron cloud and is therefore easily polarized, while a hard compound is one that has a small electron cloud and is therefore not easily polarized. Hardness value of the phenolic compounds was calculated in table 4 and ranges from 0.01 to 0.07 eV, except Rutin, which has a value of 0.0048 eV, and this means that Rutin is the softest and most active compound. The softness values of most phenolic compounds vary between 13 and 18 eV, except Rutin and Chlorogenic acid, which have the highest values. Overall, the activity of the Galangal-cinnamon spice mixture can be predicted by its softness value which was a whole number. By calculating these properties for a large number of compounds, it is possible to develop a model that can predict the biological activity of new compounds. This can be a valuable tool for drug discovery, as it can help to identify compounds that are likely to be effective against a particular target. So the activity of mixture can be predicted by the following: By collecting the experimental concentrations of each and dividing each one of them by the total, then multiplying by collecting the percentage of each component in relation to the total of 12 detected phenolic. Multiplying each of these values by its softness value to produce a valuable effectiveness, then the effectiveness of each component is divided by the total to produce the quantum relative concentration. According to the hypothesis, softness values and experimental concentrations, rutin show the highest quantum relative concentration (61.95 %) . Among other phenolic compounds detected in galangal-cinnamon aqueous extract (Table 2). According to the data presented in Table 3 that shows a high reactivity characteristic of bioactive compounds resulted from the presence of (OH) phenolic groups which have alone pair of electrons that act as nucleophile and can interact with soft acceptor molecules (electrophile), as the coronavirus protein.
The HOMO and LUMO maps of these molecules are shown in Fig. 4. The LUMO and HOMO map shows anti-bonding and bonding characteristics. From the HOMO and LUMO maps, it can be seen that the electron densities of the frontier molecular orbitals are more concentrated on six-membered ring and attached by OH groups. These regions are assumed to be chemically active, in accordance with the frontier molecular orbital theory.
Electrostatic potential (ESP) suggests the molecules’ electrophilic and nucleophilic nature and it is an important tool to study the compounds’ reactivity nature. The maps of ESP at the surface are expressed by 17 distinct colors. The blue color stands for the highest amount of the positive region where the nucleophilic reaction occurs and the reddish region indicates the negative region where the electrophilic reaction takes place while, zero potential is represented by the green color. The molecular electrostatic potential (ESP) of the bioactive molecules is shown in Figs. 4. Most of the electron density is localized on the OH phenolic groups (active site), which act as nucleophiles. The ESPs shows that the oxygen atoms of the hydroxyl group provide favorable sites for hard-hard interactions (Fig. 4) between the nucleophile and the electrophile, which are active sites for electrophilic attacks.
Rhoda Oyeladun Oyewole, Abel Kolawole Oyebamiji, and Banjo Semire., Theoretical calculations of molecular descriptors for anticancer activities of 1, 2, 3-triazole-pyrimidine derivatives against gastric cancer cell line (MGC-803): DFT, QSAR and docking approaches., Heliyon. 2020 May; 6(5): e03926.(2020). doi: 10.1016/j.heliyon.2020.e03926
In figure 1, what is the scale of the point in the horizontal axis? It doesn’t seem to be logarithmic scale; Moreover, 1-B has a (0,0) point?
Data was analyzed using Microsoft Excel® and the IC50 value was calculated using Graph pad Prism 6® by converting the concentrations to their logarithmic value and selecting non-linear inhibitor regression equation (log (inhibitor) vs. normalized response – variable slope equation).
The choice to compare chemical softness calculated by DFT with the experimental activity was interesting and does make sense. However, is not clearly discussed how the chemical softness of this phenolic compounds were related to the proposed antiviral activity.
The discussion of the chemical softness is discussed in details know
The docking calculations seemed not to play a relevant role in the article’s findings. The docking can corroborate the findings of the manuscript, but it is not clearly written.
The whole docking study is expanded and redone and the discussion is clearly written
Round 2
Reviewer 1 Report
Thank you for your revision.
I am also concerned about the protonation state of compounds in this study.
The author should check for the pKa of all compounds before QM and docking calculation.
Because the -OH group in some compounds could be -O negative at pH7.
Please cite https://www.ncbi.nlm.nih.gov/pmc/articles/PMC7825826/ and https://pubmed.ncbi.nlm.nih.gov/34109016/ in the rutin docking discussion.
Minors:
Figures 2-3: The compound’s label is missing from the revised version. Quality and resolution are still needed to improve. Please see an example from https://www.nature.com/articles/s41467-019-13973-x.
Figure 5: (Figure 4 in revised version) This is the same as the previous version. Please remake according to our comments.
Line126: elec-trons > electrons
Line139: So > So,
Line170: remove “frontier molecular orbital”
Line198: wrong format of citation
Line323: “SARS-COVID-2” > COVID19
“E Homo”, “E Lumo”, “Rutin”, “fig” are inconsistencies throughout the revised version.
Author Response
Thank you for your revision.
I am also concerned about the protonation state of compounds in this study.
The author should check for the pKa of all compounds before QM and docking calculation.
Because the -OH group in some compounds could be -O negative at pH7.
Dear Sir, Thank you for your comment. The pKa of all the structures are more than 7 so at pH of 7 they have to be fully protonated (according to table 1 attached) and that is exactly what our results show in figure 4. Accordingly, these prepared structures (fully protonated) were used for the docking step. The problem where is about using Discovery studio for visualizing the 2D interaction. Discovery studio by default only show Hydrogen atoms involved in the interaction with the protein residues or have a role in the chirality of the structure, that is high not all Hydrogen atoms are shown.
|
Chemical structure |
pKa |
|
Rutin |
7.6682944 |
|
Chlorogenic acid |
8.3453302 |
|
Quercetin |
7.2866545 |
|
Kaempferol |
7.5117359 |
|
Coffeic acid |
7.0619226 |
|
Ellagic acid |
7.332788 |
|
Coumaric acid |
7.1491294 |
|
Syringic acid |
7.5494967 |
|
Cinnamic acid |
7.2080965 |
|
Gallic acid |
7.1921763 |
|
Methyl gallate |
7.2389908 |
|
Catechin |
7.8637128 |
Attached below a figure for Rutin (2D and 3D interactions) with HCoV-229E pocket. As shown in the figure the structure is fully protonated in the 3D figure obtained by the same program (Discovery studio). But due to our large number of complexes simulated in this project we cannot supply all 3D interaction figures. But if you want we can make a supplement with 3D interaction figures or even the original pdb files.
Please cite https://www.ncbi.nlm.nih.gov/pmc/articles/PMC7825826/ and https://pubmed.ncbi.nlm.nih.gov/34109016/ in the rutin docking discussion.
The references are added
Figures 2-3: The compound’s label is missing from the revised version. Quality and resolution are still needed to improve. Please see an example from https://www.nature.com/articles/s41467-019-13973-x.
The figures were improved
Figure 5: (Figure 4 in revised version) This is the same as the previous version. Please remake according to our comments.
Attached is a re-graph of the ESP for major phenols (Rutin, Ellagic, Gallic and Cinnamic) in the main manuscript and the rest of the compounds in the Supplementary information file containing Spectrum bar. As for the spatial shape of the resulting compounds, they are derived from Gauss-view after quantum calculations, and some published researches have the exact same shapes. https://doi:10.1016/j.molstruc.2017.11.033
https://doi.org/10.1038/s41598-019-55793-5.
Line126: elec-trons > electrons
Corrected
Line139: So > So,
Corrected
Line170: remove “frontier molecular orbital”
Corrected
Line198: wrong format of citation
Corrected and added to references
Line323: “SARS-COVID-2” > COVID19
Corrected
“E Homo”, “E Lumo”, “Rutin”, “fig” are inconsistencies throughout the revised version.
Corrected throughout the manuscript

Reviewer 2 Report
The suggestions were answered by the authors.
Author Response
Thank you very much
Round 3
Reviewer 1 Report
Dear Authors,
This revised contains so many small mistakes and inconsistencies, for example,
- the text label in Fig.1
- minus sign is not appropriated
- the terms "EHOMO" and "ELOMO" are also inconsistent throughout this manuscript.
- Fig. 4 is not ready for publishing, please newly create
- Table 4 is not the correct format
- section 4.5 has a small mistake in the title
-
Author Response
- The text label in Fig.1
The labels are corrected in the figure
- minus sign is not appropriated
Corrected
- the terms "EHOMO" and "ELOMO" are also inconsistent throughout this manuscript.
Corrected throughout the manuscript
- Fig. 4 is not ready for publishing, please newly create
Newly created
- Table 4 is not the correct format
Adjusted to the correct format
- Section 4.5 has a small mistake in the title
Corrected
